# The Influence of Various Preparation Parameters on the Histological Image of Bone Tissue during Implant Bed Preparation—An In Vitro Study

Piotr Kosior [1], Piotr Kuropka [2], Maciej Janeczek [2], Marcin Mikulewicz [3], Wojciech Zakrzewski [4] and Maciej Dobrzyński [5],*

1 Department of Conservative Dentistry with Endodontics, Wroclaw Medical University, Krakowska 26, 50-425 Wroclaw, Poland; piotr.kosior@umed.wroc.pl

2 Department of Biostructure and Animal Physiology, Wroclaw University of Environmental and Life Sciences, Kozuchowska 1, 51-631 Wroclaw, Poland; piotr.kuropka@upwr.edu.pl (P.K.); maciej.janeczek@upwr.edu.pl (M.J.)

3 Department of Maxillofacial Orthopaedics and Orthodontics, Division of Facial Abnormalities, Wroclaw Medical University, Krakowska 26, 50-425 Wroclaw, Poland; marcin.mikulewicz@umed.wroc.pl

4 Department of Experimental Surgery and Biomaterials Research, Wroclaw Medical University, Bujwida 44, 50-345 Wroclaw, Poland; wojciech.zakrzewski@student.umed.wroc.pl

5 Department of Pediatric Dentistry and Preclinical Dentistry, Wroclaw Medical University, Krakowska 26, 50-425 Wroclaw, Poland

* Correspondence: maciej.dobrzynski@umed.wroc.pl; Tel.: +48-71-7840-378

**Abstract:** The purpose of this study was to present the level of bone tissue deformation after drilling under variable conditions in three different dental implant systems in a microscopic analysis. Straumann, Osstem, and S-Wide systems were used to drill boreholes in 27 porcine ribs at three different rotation speeds and under three different cooling conditions. The material was analyzed using a Nikon 80i microscope. The analysis concerned the morphological quality of the obtained boreholes. The statistical analysis revealed that satisfactory results in all drilling systems were obtained when the rotational speed did not exceed 800 revolutions per minute (rpm) regardless of the cooling temperature. However, increased rotational speed and cooling at 4 °C produced better results than without cooling in all the tested systems. Different implant systems have unique drill geometry and therefore generate differences in tissue damage under various conditions. In the experiment, a sufficient required structure was obtained in all systems, but the Straumann system yielded the best results under all the examined conditions.

**Keywords:** implant drilling; bone tissue; osseointegration

## 1. Introduction

The correct technique of bone preparation before implant placement is a fundamental prerequisite for obtaining optimal osseointegration, which is the main determinant of success in implant therapy [1–3]. An important goal in this process is to minimize mechanical and thermal trauma of bones. Increased temperature within the prepared tissue exceeding 47 °C and lasting longer than 60 s leads to permanent denaturation of the organic bone component, which could significantly disturb the osseointegration process [4–6]. This manifests itself as the cracking and disruption of the architecture of collagen fibers. As a result, the susceptibility of bone to compression is changed, and thus the stiffness of the resulting implant-to-bone-connection also changes.

One of the most common techniques concerning implant insertion is based on a conventional drilling technique. There is a gradual drilling of the osteotomy site, leading to a sequential enlargement of the drill diameter [7,8]. According to Alghamdi [9], the implant insertion using an undersized drill causes high insertion torque and results in elevated

initial stability. It is important to underline that underpreparation of an implant osteotomy site is defined as preparation of the implant's bed narrower than the implants inserted diameter [10,11], and over-preparation causes a wider implant bed preparation than the implant's inserted diameter [8]. According to O'Sullivan et al. [12], microfractures during implant insertion can happen but depend on the discrepancy between the diameter of the implant and the implant's bed, as well as bone density. Becker et al. [13] confirmed that implants placed at the time of extraction and inserted into native bone, not directly into the extraction socket, have a higher degree of initial stability. According to the literature, this stability has been demonstrated as a necessary condition required to obtain the proper osseointegration of the dental implant [14,15].

It should be noted that it has long been observed that too rigid anastomoses could lead to osteolysis. Good-quality drills and correctly selected speed should allow us to obtaining a smooth, even surface of the borehole, without leaving collagen fiber debris or ground material saturated with hydroxyapatite.

The generation of excessive temperature changes is influenced by many factors, including drill speed, the geometry of the drill surface, drill wear, and method and intensity of cooling [16–18]. This is particularly important while drilling long tunnels in bone, such as preparation for bicortical or zygomatic implants [19,20]. The shape/material of the drill is important for the subsequent healing process and implant integration. Leaving dead, carbonated or calcified tissues can affect the induction of osteogenesis or immune response. This leads to the intensification of bone resorption, which directly translates into the ability of the implant to connect to the bone [21,22]. Different drill geometries have an impact on the nature of the borehole, including the amount of debris remaining in the drilled hole and the degree of soft tissue damage.

One of the methods of implant material classification is based on the type of material used and the biologic response it causes when implanted [23]. Among the biotolerant materials, we can distinguish metals such as Gold Co-Cr alloys, Niobium, Tantalum or Stainless steel or polymers, including Polyethylene Polyamide or Polyurethane [24]. Bioinert materials examples are Titanium alloys or oxides, such as Aluminium and Zirconium oxide. Bioactive ceramic materials include Hydroxyapatites, Bioglass Carbon-silicon or Tricalcium phosphate.

Microtopography of implants plays a crucial role in effective osseointegration. It is linked to microroughness at a micrometer scale and can be modified using techniques such as acid-etching, grit-blasting or coating procedures [25]. Microtopography of implant surfaces is believed to influence the cellular level of osseointegration [26], while Nanotopography acts on cell–implant interactions at the protein and cellular levels [27,28]. Examples of implant nanotopographical surface modifications include laser ablation [29], Titanium oxide blasting [30] or anodic oxidation [31].

The present study was designed to demonstrate the diversity in the image of bone wall microstructure after preparation under a variety of conditions regarding ranges of rotational speeds and cooling intensity.

## 2. Materials and Methods

In total, 27 samples of porcine ribs were analyzed to which three different implant systems were applied (Osstem (D1), Strauman (D2), Neobiotech S-Wide (D3) (Figure 1)) at three rotational speeds: I—800, II—1200, III—1500 rpm. The boreholes were drilled under three different environmental conditions: 1—without cooling (WC), 2—with cooling using physiological saline at room temperature (CRT), 3—with cooling with normal saline at 4 °C (C4 °C). Drilling was done by the free hand method. Each drill was inserted to a depth of 10 mm. Cortical thickness of the ribs was around 1 mm, and the periosteum was entirely removed. Specimens were stored in physiological solution. All drills were used with similar parameters for comparison purposes: Pilot drill, medium diameter drill, and final implant forming drill. Each system had a similar drill sequence. The drills had a diameter of 2–3 mm. In the histological sections, we were able to determine the thickness

thanks to the pilot drillings. The anatomy of all tested ribs was similar. The periosteum was entirely removed. The ribs used for experiment were kept at room temperature and were used immediately after removal from animals. After animals were sacrificed, ribs were removed for future analysis. General macroscopic analysis was performed immediately after bone removal. To avoid chemical treatment of the material before mechanical tests, the material was kept at −20 °C. The axial pressure on the drill during preparation varied, depending on the drilling phases, from 0 to 750 gf. A hole was made for three speed ranges of the drill and three types of cooling. For the preparation procedure, drill bits were used in the sequence to prepare the bone bed for the smallest diameter of the implant for a given system (Osstem—3,0 mm, Straumann—3,3 mm, S-wide—5,5 mm).

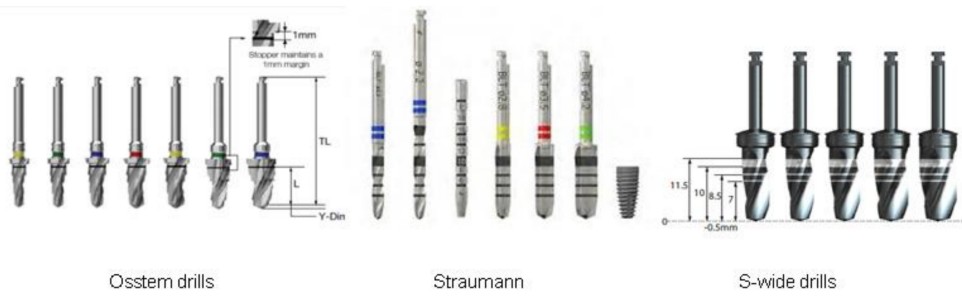

**Figure 1.** The drills used in the experiment.

The material was evaluated directly using a Nikon Eclipse 80i fluorescence microscope by the induction of autofluorescence using a UV-2A filter (Ex-330–380 nm, BA-420 nm). The material was analyzed at x40 magnification. To assess the parametric quality of the drill holes made, semi-quality assessment was used as the scoring method. The following criteria of drill hole quality assessment were used:

0  —smooth drill hole surfaces, even edges, no compression, no charring, no cracks or damage to adjacent tissues—positive—OK
1  —surface smoothness (jagged drill holes, uneven edges, presence of tissue residues)— not positive—NOK
2  —the presence of compressed (pressed) tissues—not positive—NOK
3  —presence of carbonization (char)—not positive—NOK
4  —cracking and damage to neighboring tissues—not positive—NOK

*Statistical Analysis*

Descriptive data were presented as the number of observations and fraction (%). Qualitative data were analyzed using Fisher's exact test. Odds ratio values were calculated to quantify the strength of association between satisfactory high-quality drill hole surfaces and selected variables. *p*-values <0.05 were assumed as statistically significant, and $0.05 < p < 0.1$ were interpreted as a tendency toward a statistically significant value. Statistical analysis was performed using Statistica v. 13.3 (Tibco Software Inc., Palo Alto CA, USA).

## 3. Results

*3.1. Osstem (D1)*

In this drill system, a smooth surface is visible and there is no damage to surrounding tissues (Figure 2) in the WC group. The bone structure just below the edge of the drill hole at speeds I and II remains unchanged. At −1500 rpm, small debris remains while the bone tissue is subject to partial compression. Signs of mild carbonization are present on the surface at speeds II and III. At 800 rpm, the surface is compact but uneven and slightly frayed. This is especially visible at 1200 rpm (Figure 2d—arrow). Bones adjacent to the border do not show structural disturbances. At 1500 rpm, slight carbonization (arrow) is visible (Figure 2f).

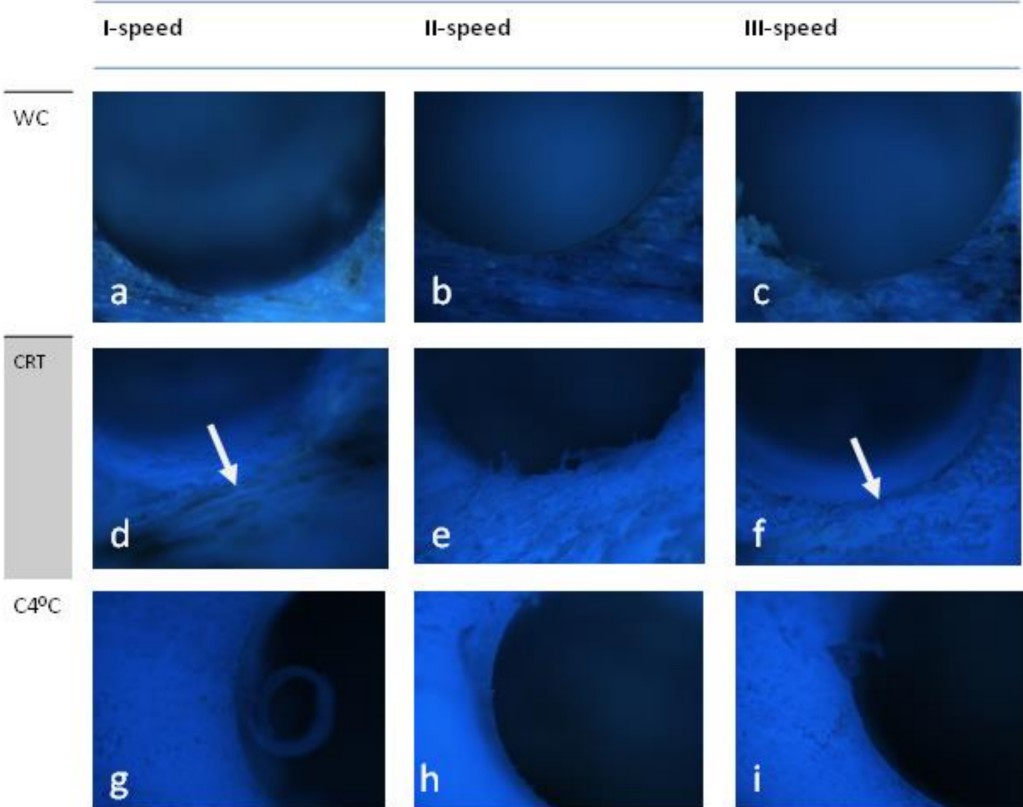

**Figure 2.** Drill-Osstem. The surfaces of holes drilled at different speeds (I-800 rpm, II-1000 rpm, III-1200 rpm) and using different cooling methods. Note the different smoothness and different damage in the adjacent areas. WC-without cooling, CRT-cooling at room temperature, C4 °C cooling at 4 °C. Autofluorescence. Mag. x40. (**a–c**) WC-without cooling, (**d–f**) CRT-cooling at room temperature, (**g–i**) C4 °C cooling at 4 °C.

Cooling at 4 °C. Smooth surface at all rotation speeds. No signs of carbonization. They stay on the surface (especially at 800 rpm), and there are small chips merged with the bone (arrow). At 1200 rpm, bone fragments at the border are slightly compressed (arrow). At 1500 rpm, the borehole is even, with no signs of damage to adjacent tissues.

### 3.2. Straumann (D2)

In this drill system, without cooling (WC), a smooth surface is observed at lower speeds (1 and 2) (Figure 3a,b). At 1500 rpm, uneven drill holes are visible (Figure 3c). No signs of carbonization. CRT—a very smooth surface at speeds I and II. At speeds II and III, small chips remain with signs of carbonization. C4 °C—at all speeds, characteristic chips with signs of carbonization remain (Figure 3g–i—arrow). Adjacent tissue without damage.

### 3.3. S-Wide (Neobiotech S-Wide)

WC—slightly uneven but smooth surface at all speeds without damage to adjacent tissues. At room temperature at speeds I and II, small chips with signs of carbonization remain on the surface (Figure 3b-arrow). At 1500 rpm, the surface is smooth. No changes in adjacent tissues. At all speeds, small chips with signs of carbonization (arrow) remain. At 800 rpm they are fragmented, while at speeds II and III they form a distinct line. The surface is smooth and adjacent tissues are unchanged (Figure 4d–i).

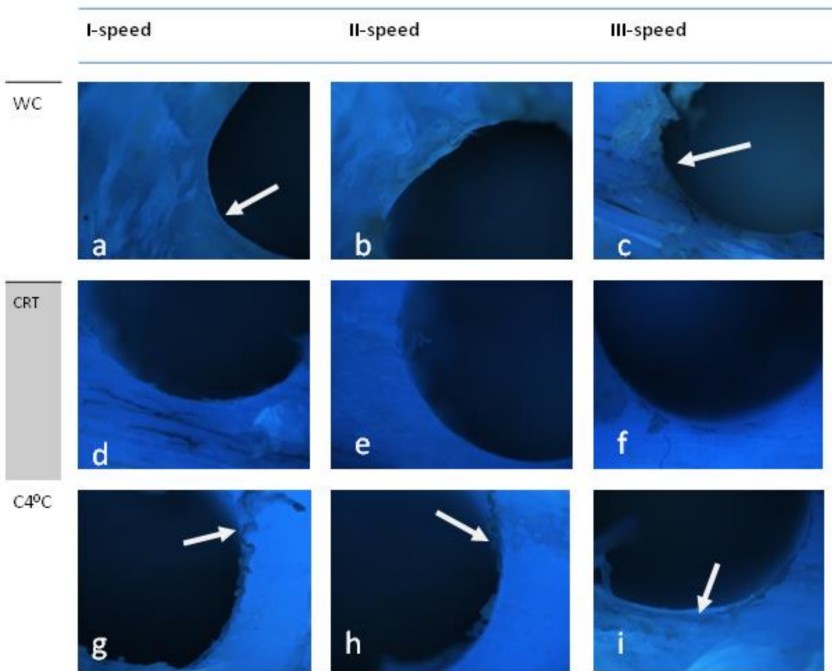

**Figure 3.** Drill-Straumann. The surfaces of holes drilled at different speeds (I-800 rpm, II-1000 rpm, III-1200 rpm) and using different cooling methods. Note the different smoothness and signs of carbonization(arrow). WC-without cooling, CRT-cooling at room temperature, C4 °C cooling at 4 °C. Autofluorescence. Mag. x40. (**a–c**) WC-without cooling, (**d–f**) CRT-cooling at room temperature, (**g–i**) C4 °C cooling at 4 °C.

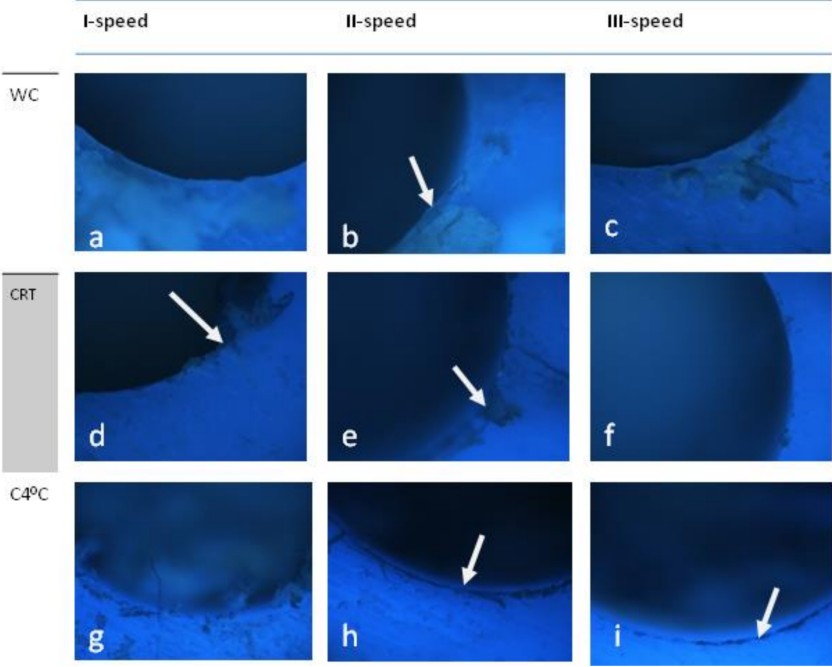

**Figure 4.** Drill-S-wide. The surfaces of holes drilled at different speeds (I-800 rpm, II-1000 rpm, III-1200 rpm) and using different cooling methods. Note the different smoothness and signs of carbonization (arrow). WC-without cooling, CRT-cooling at room temperature, C4 °C cooling at 4 °C. Autofluorescence. Mag. x40. (**a–c**) WC-without cooling, (**d–f**) CRT-cooling at room temperature, (**g–i**) C4 °C cooling at 4 °C.

## 4. Scoring

The results of the quality assessment of the holes in the tested material ranged from 0 to 2 points. It was assumed that a score of 0 is a satisfactory value (OK) and scores of 1 and 2 are not acceptable (NOK) (Table 1).

**Table 1.** Results of scoring in microscopic evaluation.

| $X_1$ Systems | $X_2$ $v_c$ (m/s) | $X_3$ Environmental Conditions | Quality of the Drill Hole Surface (Score) | $Y$ |
|---|---|---|---|---|
| 1—Osstem | 0.147 | 1—without cooling | 0 | OK |
| 1—Osstem | 0.220 | 1—without cooling | 0 | OK |
| 1—Osstem | 0.275 | 1—without cooling | 1 | NOK |
| 1—Osstem | 0.147 | 2—with cooling at room temperature | 2 | NOK |
| 1—Osstem | 0.220 | 2—with cooling at room temperature | 2 | NOK |
| 1—Osstem | 0.275 | 2—with cooling at room temperature | 2 | NOK |
| 1—Osstem | 0.147 | 3—with cooling with normal saline at 4°C | 1 | NOK |
| 1—Osstem | 0.220 | 3—with cooling with normal saline at 4°C | 1 | NOK |
| 1—Osstem | 0.275 | 3—with cooling with normal saline at 4°C | 1 | NOK |
| 2—Strauman | 0.147 | 1—without cooling | 0 | OK |
| 2—Strauman | 0.220 | 1—without cooling | 0 | OK |
| 2—Strauman | 0.275 | 1—without cooling | 1 | NOK |
| 2—Strauman | 0.147 | 2—with cooling at room temperature | 0 | OK |
| 2—Strauman | 0.220 | 2—with cooling at room temperature | 0 | OK |
| 2—Strauman | 0.275 | 2—with cooling at room temperature | 1 | NOK |
| 2—Strauman | 0.147 | 3—with cooling with normal saline at 4°C | 1 | NOK |
| 2—Strauman | 0.220 | 3—with cooling with normal saline at 4°C | 1 | NOK |
| 2—Strauman | 0.275 | 3—with cooling with normal saline at 4°C | 1 | NOK |
| 3—Neobiotech S-Wide | 0.230 | 1—without cooling | 1 | NOK |
| 3—Neobiotech S-Wide | 0.346 | 1—without cooling | 1 | NOK |
| 3—Neobiotech S-Wide | 0.432 | 1—without cooling | 1 | NOK |
| 3—Neobiotech S-Wide | 0.230 | 2—with cooling at room temperature | 1 | NOK |
| 3—Neobiotech S-Wide | 0.346 | 2—with cooling at room temperature | 1 | NOK |
| 3—Neobiotech S-Wide | 0.432 | 2—with cooling at room temperature | 1 | NOK |
| 3—Neobiotech S-Wide | 0.230 | 3—with cooling with normal saline at 4°C | 1 | NOK |
| 3—Neobiotech S-Wide | 0.346 | 3—with cooling with normal saline at 4°C | 1 | NOK |
| 3—Neobiotech S-Wide | 0.432 | 3—with cooling with normal saline at 4°C | 1 | NOK |

We observed significantly higher frequencies for drilling at a cutting speed of no more than 0.22 m/s than at a speed higher than 0.22 m/s ($p = 0.003$) (Table 2). The type of system and cooling method are at the threshold of statistical significance ($p = 0.076 > 0.05$). The authors assumed that drilling with cutting speeds up to 0.22 m/s with the Straumann system without cooling or with liquid cooling at ambient temperature yields the best results.

**Table 2.** Distribution of study preparation parameters in the groups differing in the quality of the drill hole surface.

| Parameters | The Quality of the Drill Hole Surface | | | | Fischer's Exact Test *p*-Value |
|---|---|---|---|---|---|
| | OK | | NOK | | |
| | *n* | % | *n* | % | |
| System: | | | | | |
| Osstem | 2 | 33.3% | 7 | 33.3% | 0.076 * |
| Straumann | 4 | 66.7% | 5 | 23.8% | |
| Neobiotech S-Wide | 0 | 0.0% | 9 | 42.9% | |
| Environmental conditions: | | | | | |
| Without cooling | 4 | 66.7% | 5 | 23.8% | 0.076 * |
| With cooling at room temperature | 2 | 33.3% | 7 | 33.3% | |
| With cooling with normal saline at 4 °C | 0 | 0.0% | 9 | 42.9% | |
| Cutting speed $v_c$ (m/s) | | | | | |
| <0.22 | 6 | 100.0% | 6 | 28.6% | 0.003 ** |
| >0.22 | 0 | 0.0% | 15 | 71.4% | |

*: tendency to a statistically significant value; **: statistically significant.

The probability of obtaining a high-quality drill hole (OK) when machining at a cutting speed of no more than 0.22 m/s is 100%, and all results are positive for this option. The probability of obtaining a high-quality drill hole using the Straumann system or when drilling without cooling is six times higher (OR = 6.4), but the 95% confidence interval includes the value of 1, which means that the chances for very good quality of the drill hole (OK) using the Straumann system or drilling without cooling are the same as when using any of the other systems or when drilling with cooling (Table 3).

**Table 3.** The likelihood of a high-quality drill hole surface occurrence in relation to the used system and cooling/non-cooling condition.

| System and Cooling | High Quality | | OR [95% CI] |
|---|---|---|---|
| | OK | NOK | |
| For: Straumann | 4 | 5 | 6.40 [0.89–46.00] |
| For: Osstem or Neobiotech S-Wide | 2 | 16 | 0.16 [0.02–1.12] |
| For: without cooling | 4 | 5 | 6.40 [0.89–46.00] |
| For: with cooling | 2 | 16 | 0.16 [0.02–1.12] |

OR—odds ratio, CI—confidence interval.

## 5. Discussion

Rotational speed builds up the momentum of the cutting edge of the drill to cut small homogeneous fragments of tissue without causing too deep damage to the surrounding tissue. Good drills should have this momentum at the speed range of approx. 1200 rpm because it is considered optimal for bone tissue [32]. Lower rotational speeds may be insufficient to uniformly remove tissues of a particular thickness. On the other hand, too high a rotational speed of the drill can generate excessive heat damage to the adjacent bone [32]. To prevent this, cooling solutions are used to remove excess heat. Because the fluids in the experiment could have been in contact with the drill only on its outer surface, the cooling efficiency, in this case, should be related to the ability of the drill to evenly distribute heat throughout its mass.

The obtained results indicate that cutting speed has a direct impact on the drill hole wall structure. In the experiment carried out, the Osstem drill was characterized by good cutting properties at all speeds and cooling methods, except for speeds I and II when cooled at room temperature. Under these conditions, minor disturbances in bone structure appeared in the form of jagged fibers. In this case, it means that drilling under such

conditions should take place at higher speeds because at 1500 rpm, this effect was not observed. Similar results were obtained by Yeinol et al. [32].

The Straumann drill was characterized by very good parameters at all rotational speeds and cooling methods. Small residues of bone chips were observed at speeds I and II with cooling at 4 °C. This can be related to the shape of the working part of the drill and the materials used that evenly distribute heat.

In the case of the Neobiotech S-Wide application, slightly greater destruction of adjacent tissues and tissue carbonation were observed for the same conditions. In addition, areas of compression and distortion of adjacent tissues were visible.

Initial mechanical engagement in the healing phase is important for the remodeling process [33–36]. The osseointegration process results in achieving secondary stability, which is a direct contact between the intact bone and implant surface [21,22,37].

In the study, all drills allowed us to obtain the desired hole characteristics; all drills require adequate rotations and an appropriate cooling temperature.

However, in some cases (D2), the cooling temperature was not sufficient to prevent tissue carbonization. We observed carbonization signs under water cooling and at 4 °C, but not at room temperature. This problem perhaps is associated with the shape of drill and the resistance of water flowing back, which results in locally increased friction and a temperature rise, leading to carbonization. However, in this case, this is not a very significant problem, which should be considered, but as it occurs too intensively, may be a factor causing carbonized residues to remain in the drilled hole. Such remains may be the cause of non-specific inflammation, which is not recommended for dental implantation.

Dangerous effects of excessive rotational speed can be observed in the case of drills with a larger diameter due to the effect of drill geometry (effect of drill radius on tangential speed). This entails the risk that after the caliber of the drill hole has been determined, the inserted implant will not achieve the desired primary stability. Insufficient implant stability could hamper the osteoconductive capacity of the implant [38], resulting in impaired healing and implant loss [39]. Many factors could affect primary stability, including bone quantity and quality, implant macro- and micro-geometry, and surgical technique [40].

On the other hand, the obtained drill hole may be too small, for example due to leaving a part of the material mineralized in the hole. Undersized drilling is one of the most common surgical procedures to improve implant primary stability. With this technique, the implant osteotomy diameter is substantially smaller than the diameter of the implant [41]. It has been reported that increased lateral bone compression during implant placement into an undersized site results in higher insertion torque (ITQ), which can be an indicator of improved primary stability [10].

Elevated lateral compression during preparation can also be dangerous. However, a recent systematic review [42] suggests that higher lateral bone compression does not always result in enhanced mechanical stability and enhanced osseointegration. One of studies pointed out that implant insertion in underprepared sites provokes ischemia, osteocyte necrosis, and generates micro-fractures in the bone. It is well known that damaged cortical bone is replaced in the remodeling process. In the resorption process, temporary pores are formed in the peri-implant bone known as the remodeling space [2]. Based on our own observations and those described in the literature, the aim of drilling an appropriate hole is to obtain the so-called maximum primary stability followed by its temporary weakening. At this point, rapid bone resorption occurs, and bone density decreases directly in the area of the implants, which is followed by the synthesis of new bone tissue, creating a permanent biomechanical connection. Then, a slow process of bone tissue remodeling will last for many years. Remodeling occurs at the bone–implant interface, with the bone implant presumably affected by static strain. However, the extent of the remodeling has never been assessed by comparing different drilling protocols using implants with the same macro- and micro-design. It could be assumed that an insignificant amount of cortical bone could be subject to substantial compression forces. Regardless of the obtained results, in vitro studies will only become more valid predictors of human reactions to exposures

and treatments if there is substantial improvement in both their scientific methods as well as in more systematic reviews of the animal or "in vitro" literature as it evolves. Systematic reviews of animal or "in vitro" research, if used to inform the design of clinical trials, particularly with respect to appropriate method timing and other crucial aspects of the experiment, will further improve the predictability of animal research in human clinical trials.

The intensity of this process depends on many individual features, including the degree of blood supply, the activity of immune cells, and bone marrow, because the cells responsible for osteolysis will be supplied in these processes. The synthesis of the new bone will be associated with an appropriate hormonal and vitamin status (Vit. D3, K, E) and also with the availability of the correct proportions of calcium and phosphorus. The results suggest that the drilling technique has a significant impact on both the interfacial cellular reactions and on the circumferential peri-implant bone tissue.

## 6. Conclusions

The obtained results include a diverse amount of bone deformations and damage after preparation with drills. Undoubtedly, the greater number of necrotic deformities will proportionally disturb the osseointegration process more due to the necessity of their resorption and reconstruction into living tissue. This occurs in the presence of superficial inflammation that reduces the primary stability of the implant. Primary stability of the implant is compromised due to the appearance of superficial inflammation of the tissue.

The optimal bone surface structure after implant preparation should be characterized by the absence of necrotic or carbonated zones and a structure that is not too dense to allow free vessel proliferation. Such features are characteristic of the Straumann system while the other drills were slightly less effective.

**Author Contributions:** Conceptualization P.K. (Piotr Kosior), M.J. and M.D.; methodology, P.K. (Piotr Kosior), P.K. (Piotr Kuropka) and M.D.; software, W.Z. and M.M.; validation, P.K. (Piotr Kosior), P.K. (Piotr Kuropka) and M.D.; formal analysis, M.J. and M.M.; investigation, P.K. (Piotr Kosior), P.K. (Piotr Kuropka) and M.D.; resources, P.K. (Piotr Kosior), P.K. (Piotr Kuropka) and M.D.; data curation, P.K. (Piotr Kosior), P.K. (Piotr Kuropka), W.Z. and M.D.; writing—original draft preparation, P.K. (Piotr Kosior), P.K. (Piotr Kuropka) and M.D.; writing—review and editing, P.K. (Piotr Kosior), W.Z. and M.M.; visualization, W.Z.; supervision, M.J. and M.M.; project administration, P.K. (Piotr Kosior) and M.D.; funding acquisition, M.J. All authors have read and agreed to the published version of the manuscript.

**Funding:** This research received no external funding.

**Institutional Review Board Statement:** Not applicable.

**Informed Consent Statement:** Not applicable.

**Conflicts of Interest:** The authors declare no conflict of interest.

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
