# Peer review of "The Influence of Various Preparation Parameters on the Histological Image of Bone Tissue during Implant Bed Preparation—An In Vitro Study"

_applsci, doi:10.3390/app11041916_

Round 1

Reviewer 1 Report

The paper describes experiences with implant bed preparations. The study is not novel, but might contain new and usefulk informations for the readers.

I think the methods part should be significantly improved, and unfortunatelly I suppose some basic deficiency in the study design as well.

1) The drilling standardization, or the parameters of drillings were not given in details. What was the method for drillings? Free hand? Drilling device or some "drilling tower"? How was the drilling's perpendicular direction secured? What was the axial pressure during drillings, and how was it standardized? I think this is a crucial point in terms of heat development and cavity injuries as well. The drilling lenght was not given, only the " smallest diameter of implant for a given system".

2) Animal specimens. How many animals gave the 27 ribs, how old and which gender animals? Was the periosteum entirely removed? Did you measure cortical thicknesses of the ribs? How was rib stored (dry, in phys. salt?, cooled? frozen?)? These can strongly influence results.

3) My biggest concern is, that e.g. by the Straumann system the manufacturer's protocol was not kept. This surgical protocoll is very important before implant insertion, even according to my personal experience with this system. I mean, the guide prescribes to use as the final drill, the so called 'profile drill' in hard and very hard bones. According to my earlier works, porcine ribs simulate usually D1, D2 mandible bones with usually 2.0-2.4mm cortical thicknesses. In such a bone, if you do not use lastly the 'profile drill' with 300 rpm, it is a surgical procedural mistake. Differences of implant systems and drills can be usefull for clinicians, if drillings are according to the prescribed protocols of the manufacturers in my opinion.

4) You writes in the methods, 800 rpm, 1200 rpm, 1500 rpm drilling speeds. In the results you writes about drilling speeds of 0.147-0.432 m/s (table 1 and 2). This is very confusing. In m/s readers would expect something with feeding rates (i.e., result of cumulative effect of axial pressure and rotational speed). I suppose, however, these mentioned speeds are rather revolutional/rotational speeds.

5) In figures 2-4 the concrete drilling (rpm) speeds should be given, rather than speed I-II-III. Easier for the readers. In these figure legend descriptions you could specify which system is that above.

6) Arrows in figures 2-4 indicates carbonization according to your legend. Later in the discussion readers do not get answers, why coolant at 4°C results in impaired results, similar to WC by (fig 3 h,i,j)? 

Author Response

Dear Reviewer,

We would like to express our sincerest gratitude to the Reviewers for their enormous efforts in criticizing the manuscript. We have taken into account all raised question here follows the detailed answers to the Reviewers.

REVIEWER#1

- The drilling standardization, or the parameters of drillings were not given in details. What was the method for drillings? Free hand? Drilling device or some "drilling tower"? How was the drilling's perpendicular direction secured? What was the axial pressure during drillings, and how was it standardized? I think this is a crucial point in terms of heat development and cavity injuries as well. The drilling lenght was not given- only the " smallest diameter of implant for a given system".

Answer: We would like to thank you for the comment. References have been extended, Introduction section has been extended. Drills were drilled using a free hand method. Each drill was inserted to a depth of 10mm. The nature of the load was similar in all measurements. The axial pressure was from 0 to 720 g (750 ± 12 g).

- Animal specimens. How many animals gave the 27 ribs, how old and which gender animals? Was the periosteum entirely removed? Did you measure cortical thicknesses of the ribs? How was rib stored (dry, in phys. salt?, cooled? frozen?)?- These can strongly influence results.

Answer: We would like to thank you for the comment. References have been extended, Introduction section has been extended. The periosteum was entirely removed. Cortical thickness of the ribs was around 1mm. The ribs used for experiment were kept at room temperature, and were used immediately after transportation from animals. After animals sacrificing, ribs were taken for future analysis. General macroscopic analysis was performed just after bone removal. To avoid chemical treatment of material before mechanical tests, material was kept in −20 °C. Due to great similarity in bone structure of pigs ribs we took 10 ribs from each side from 2 male pigs (20 from one pig and 7 from another), 6 month old (approx 100kg).

- My biggest concern is, that e.g. by the Straumann system the manufacturer's protocol was not kept. This surgical protocoll is very important before implant insertion, even according to my personal experience with this system. I mean, the guide prescribes to use as the final drill, the so called 'profile drill' in hard and very hard bones. According to my earlier works, porcine ribs simulate usually D1, D2 mandible bones with usually 2.0-2.4mm cortical thicknesses. In such a bone, if you do not use lastly the 'profile drill' with 300 rpm, it is a surgical procedural mistake. Differences of implant systems and drills can be usefull for clinicians, if drillings are according to the prescribed protocols of the manufacturers in my opinion.

Answer: We would like to thank you for the comment. References have been extended, Introduction section has been extended. All drills were used with similar parameters for comparison purposes. Pilot drill, medium diameter drill, and final implant forming drill. Each system had a similar drill sequence. The drills had a diameter of 2-3mm. The ribs used for this scientific work had 1mm cortical bone thickness. In the histological sections, we were able to determine the thickness thanks to the pilot drillings. The anatomy of all tested ribs was similar.

- You writes in the methods, 800 rpm, 1200 rpm, 1500 rpm drilling speeds. In the results you writes about drilling speeds of 0.147-0.432 m/s (table 1 and 2). This is very confusing. In m/s readers would expect something with feeding rates (i.e., result of cumulative effect of axial pressure and rotational speed). I suppose, however, these mentioned speeds are rather revolutional/rotational speeds.  

Answer: We would like to thank you for the comment. References have been extended, Introduction section has been extended. The reference point is the value specified in the table. The units used to define the rotational speed values ​​are listed in Table 1.

- In figures 2-4 the concrete drilling (rpm) speeds should be given, rather than speed I-II-III. Easier for the readers. In these figure legend descriptions you could specify which system is that above

Answer: We would like to thank you for the comment. References have been extended, Introduction section has been extended. We would like to thank you for the comment. The speed values were added to the descriptions of aforementioned figures and were: I-800rpm, II- 1000rpm, III- 1200rpm.

- Arrows in figures 2-4 indicates carbonization according to your legend. Later in the discussion readers do not get answers, why coolant at 4°C results in impaired results, similar to WC by (fig 3 h,i,j)? 

Answer: We would like to thank you for the comment. The explanations were added to a text.

Reviewer 2 Report

The manuscript topic is actual and the paper has merit. It could be attractive, adequate and interesting for the APPLIED SCINECS journal readers. However there are some points that authors should address in order to have a final more complete paper. Authors should underline the limitation of the value of the study, and the clinical and surgical implication of the presented study should be added. At this stage the paper seems to be directed to researchers and not surgeons. Please emphasize the clinical application of the study, and its scientific rationale.

The limitation of an "in vitro study" should be underlined and need to be synthesized in a paragraph.
....in vitro studies will only become more valid predictors of human reactions to exposures and treatments if there is substantial improvement in both their scientific methods as well as in more systematic review of the animal or "in vitro" literature as it evolves. Systematic reviews of animal or "in vitro" research, if they are used to inform the design of clinical trials, particularly with respect to appropriate drug dose, timing and other crucial aspects of the drug regimen, will further improve the predictability of animal research in human clinical trials....

References are inadequate. Introduction section is poor. Some more references about the bone healing and the dental implant shape and materials as factors that may influence and conditions the final osteointegration have to be added.

Preparing a hole for a dental implant placement cannot be the same concept of placing a screw in the wall. Authors should emphasize the CLINICAL reflection of the study, and how a no correct preparation of the site could be related with final no osteointegration of the screw. 

In the discussion section authors should compare the results of the present study with others one presented and published in the literature.

Comuzzi, L.; Tumedei, M.; Piattelli, A.; Iezzi, G. Osseodensification Drilling vs. Standard Protocol of Implant Site Preparation: An In Vitro Study on Polyurethane Foam Sheets. Prosthesis 2020, 2, 76-86.

W. Nicholson, J. Titanium Alloys for Dental Implants: A Review. Prosthesis 2020, 2, 100-116.

Author Response

Dear Reviewer,

We would like to express our sincerest gratitude to the Reviewers for their enormous efforts in criticizing the manuscript. We have taken into account all raised question here follows the detailed answers to the Reviewers.

REVIEWER#2

- The manuscript topic is actual and the paper has merit. It could be attractive, adequate and interesting for the APPLIED SCINECS journal readers. However there are some points that authors should address in order to have a final more complete paper. Authors should underline the limitation of the value of the study, and the clinical and surgical implication of the presented study should be added. At this stage the paper seems to be directed to researchers and not surgeons. Please emphasize the clinical application of the study, and its scientific rationale.

Answer: We would like to thank you for the comment. References have been extended, Introduction section has been extended. The obtained results include a diverse amount of bone deformations and damage after preparation with drills. Undoubtedly, the greater number of necrotic deformities will proportionally disturb the osseointegration process more due to the necessity of their resorption and reconstruction into living tissue. This is with the presence of superficial inflammation that reduces the primary stability of the implant. Primary stability of the implant is compromised due to the appearance of superficial inflammation of the tissue.

- The limitation of an "in vitro study" should be underlined and need to be synthesized in a paragraph.

....in vitro studies will only become more valid predictors of human reactions to exposures and treatments if there is substantial improvement in both their scientific methods as well as in more systematic review of the animal or "in vitro" literature as it evolves. Systematic reviews of animal or "in vitro" research, if they are used to inform the design of clinical trials, particularly with respect to appropriate drug dose, timing and other crucial aspects of the drug regimen, will further improve the predictability of animal research in human clinical trials....

Answer: We would like to thank you for the comment. Corrections were added to the text.

- References are inadequate. Introduction section is poor. Some more references about the bone healing and the dental implant shape and materialsas factors that may influence and conditions the final osteointegration have to be added.

Answer: We would like to thank you for the comment. References have been extended, Introduction section has been extended.

- Preparing a hole for a dental implant placement cannot be the same concept of placing a screw in the wall. Authors should emphasize the CLINICAL reflection of the study, and how a no correct preparation of the site could be related with final no osteointegration of the screw. In the discussion section authors should compare the results of the present study with others one presented and published in the literature.

Comuzzi, L.; Tumedei, M.; Piattelli, A.; Iezzi, G. Osseodensification Drilling vs. Standard Protocol of Implant Site Preparation: An In Vitro Study on Polyurethane Foam Sheets. Prosthesis 20202, 76-86.

  1. Nicholson, J. Titanium Alloys for Dental Implants: A Review. Prosthesis20202, 100-116.

Answer: We would like to thank you for the comment. References have been extended, Introduction section has been extended. From the histological point of view, Osseointegration is a multi-stage process which may influence many factors. Undoubtedly, one of the most important factors is the condition of the bone tissue at the time of implantation. Preparation with rotary instruments is a traumatic process for bones, causing the formation of an open wound and the cascade of repair processes. Their intensity determines the quality of osseointegration. Thus, the degree of bone traumatization during the preparation of the implant bed has a direct impact on the tissue. Both the degree of thermal tissue damage during preparation and their macroscopic architecture will determine the course of bone healing after implantation. The primary stabilization, which is undoubtedly a preliminary determinant of osteointegration, is influenced by many factors, including, in addition to the tissue architecture, the type of implant used, its surface character and the strength with which it was embedded in the bone. The article mentioned by the reviewer has been added to the manuscript.

Reviewer 3 Report

General comments:

I am not a statistician and I could be wrong, but I have some issues with applied statistical analyses, which I would like to be clarified:

  1. If I am correct, your experimental design was without repetitions, and only one hole per drill/drilling condition was made and analyzed? Why?
  2. Why statistical analyses was performed for RMP converted for vc? Is it a tangential speed ? Tangential speed is directly proportional to r, there is no information about drills diameters (were they equal for all drills, see comment to L71). Why vc=0.22 was used as a threshold value ?
  3. What was the criteria for chi-square and Fisher’s exact test selection ? In your data, some expected values are less than 5, while for chi-square test it is assumed that all expected values are greater than 5. If not, you should use Fisher’s exact test - I agree, that generally Fisher’s exact test is used for 2x2 table, however it can be also applied for 3x2 tables (as Freeman-Halton extension ). For your data, for the first two tables where expected values are below 5, Fisher’s exact test gives p=0.116.
  4. For chi-square test, value of chi-square=6.7 is for Likelihood Ratio (LR) Chi-Square test, not Pearson's Chi-square test (which gives chi-square=5.14 and p=0.076). For LR chi-square=6.70 and p=0.035. Did you combined the results of both test? What test was applied exactly? Fisher’s or LR? Please verify
  5. Table 3. How OR was calculated when odds of event in the second group is equal 0 ? Using Haldane-Anscombe correction the OR value is 16 [1,64; 155,8]. Did you applied other correction? Which one ?

Minor comments:

Manuscript paper requires English editing and should be proofread by editor with expertise in the field. I will point out some of the examples below (not all of them).

L30 Please correct to different or unique. All system have original geometry.

L31-32 Please consider changing to good or sufficient  - “the best “ should be reserved only for .. the best

L50 “degree of wear” – please change to “drill wear” (without degree)

L63-67 Please give the detailed information about these drilling system (names of producers, ect). How does the applied rotational speeds relate to the typical drilling speed or specific drilling system operating recommendations ?

L65, L89, and others: Please unify speed I, II, III, or 1, 2, 3 (Arabic, roman)

L66 environmental condition 2 – the applied cooling medium was … (water?, phs?)

L71 that was this “the smallest diameter”? I suppose that it was different for each system. Please specify

L73 Please correct “mm” to “nm”

L76 Any reference for this scoring system? Who performed the evaluation? Was is blinded?

Figures –For each figure caption please add the drill name, explain the abreactions (cooling methods), speeds and the explanation of arrows - figures captions should be self-explanatory

L130 Please correct the table caption (title)

L132,133 Please consider changing to “contingency table” and “chi-squared test for independence”

Figure 5 repeats the results from Table 2. One should be removed.

Throughout the whole discussion, please give the exact drill names and analysed rpm, not drill D2 and speed 3.

L164 Please rephase - Results are analyzed and presented in vc, not in terms of rpm.

L172 adjoining? Did you mean adjacent ?

L175 vital bone? Did you mean intact?

L179-180 not clear. You mean geometric growth (exponential growth) or the effect of drill geometry (effect of drill radius on tangential speed) ? Please rephase.

L189-190 Neither hole diameter nor insertion torque of sample implant was measured. Please rephase or remove. Moreover, as you stated, undersized drilling is an intentional drilling for improving implant stability.

L191 strong ?

L191-192 Reference for this review needed.

L200 underway for many years? Did you mean “will last for many years” ?

L204 Which process? Compression? Please rephase.

L213 Please change “melted zones” to “carbonated areas”

L213 compact? Did you mean dense?

L212-214 Presence of osteocytes was not measured, please rephase.

Author Response

Dear Reviewer,

We would like to express our sincerest gratitude to the Reviewers for their enormous efforts in criticizing the manuscript. We have taken into account all raised question here follows the detailed answers to the Reviewers.

REVIEWER#3

- If I am correct, your experimental design was without repetitions, and only one hole per drill/drilling condition was made and analyzed? Why?

Answer: We would like to thank you for the comment. A hole was made for three speed ranges of the drill and three types of cooling.

- Why statistical analyses was performed for RMP converted for vc? Is it a tangential speed ? Tangential speed is directly proportional to r, there is no information about drills diameters (were they equal for all drills, see comment to L71).

Answer: We would like to thank you for the comment. Because the vc better describes the movement between drill and surrounding tissues. As different drill sizes are used, it is not the rotational speed itself that is more important, but the speed with which it hits and interacts with bone tissue. Yes, it is tangential speed. It was calculated according to drill diameter. Because, in course of hole preparation different sizes of drills were used (from smallest to the largest, we used for calculation the final drill diameter). The information was added to the text.

- Why vc=0.22 was used as a threshold value?

Answer: We would like to thank you for the comment. This value is a result of statistical analysis. This value is threshold resulting from independence test for all drills type . The probability of obtaining a high-quality drill hole (OK) when machining at a cutting speed of no more than 0.22 m/s is over thirty times higher than at a speed higher than 0.22 m/s (OR = 31.0, 95% confidence interval from 1.51 to 634). The information is in the result section.

- What was the criteria for chi-square and Fisher’s exact test selection ? In your data, some expected values are less than 5, while for chi-square test it is assumed that all expected values are greater than 5. If not, you should use Fisher’s exact test - I agree, that generally Fisher’s exact test is used for 2x2 table, however it can be also applied for 3x2 tables (as Freeman-Halton extension ). For your data, for the first two tables where expected values are below 5, Fisher’s exact test gives p=0.116.

Answer: We would like to thank you for the comment. This was an analysis of non parametrical data, therefore values were multiplied by 10.

- For chi-square test, value of chi-square=6.7 is for Likelihood Ratio (LR) Chi-Square test, not Pearson's Chi-square test (which gives chi-square=5.14 and p=0.076). For LR chi-square=6.70 and p=0.035. Did you combined the results of both test? What test was applied exactly? Fisher’s or LR? Please verify.

Answer: We would like to thank you for the comment. Fisher test was applied.  The results of both tests were combined.

- Table 3. How OR was calculated when odds of event in the second group is equal 0 ? Using Haldane-Anscombe correction the OR value is 16 [1,64; 155,8]. Did you applied other correction? Which one ?

Answer: We would like to thank you for the comment. The data equal 0 were removed from the test.

- Manuscript paper requires English editing and should be proofread by editor with expertise in the field. I will point out some of the examples below (not all of them).

Answer: We would like to thank you for the comment. The manuscript has been checked and corrected by an editor with expertise in the field.

- L30 Please correct to different or unique. All system have original geometry.

Answer: We would like to thank you for the comment. The manuscript has been corrected.

- L31-32 Please consider changing to good or sufficient  - “the best “ should be reserved only for .. the best

Answer: We would like to thank you for the comment. The manuscript has been corrected.

- L50 “degree of wear” – please change to “drill wear” (without degree)

Answer: We would like to thank you for the comment. The manuscript has been corrected.

- L63-67 Please give the detailed information about these drilling system (names of producers, ect). How does the applied rotational speeds relate to the typical drilling speed or specific drilling system operating recommendations?

Answer: The drilling speeds used in the experiment are the average, most frequently recommended, speed ranges for implant drills.

- L65, L89, and others: Please unify speed I, II, III, or 1, 2, 3 (Arabic, roman).

Answer: We would like to thank you for the comment. The manuscript has been corrected.

- L66 environmental condition 2 – the applied cooling medium was … (water?, phs?)?

Answer: We would like to thank you for the comment. The manuscript has been corrected. Applied cooling medium was phs.

- L71 that was this “the smallest diameter”? I suppose that it was different for each system. Please specify

Answer: For the preparation procedure, drill bits were used in the sequence to prepare the bone bed for the smallest diameter of implant for a given system ( Osstem - 3,0 mm,  Straumann - 3,3 mm, S-wide - 5,5 mm).

- L73 Please correct “mm” to “nm”.

Answer: We would like to thank you for the comment. The manuscript has been corrected.

- L76 Any reference for this scoring system? Who performed the evaluation? Was is blinded?

Answer: We would like to thank you for the comment. Any references for scoring system. The scoring system was established basing upon needs, declared by drill producers.

- Figures –For each figure caption please add the drill name, explain the abreactions (cooling methods), speeds and the explanation of arrows - figures captions should be self-explanatory.

Answer: We would like to thank you for the comment. Corrections were made in the text.

- L130 Please correct the table caption (title).

Answer: We would like to thank you for the comment. Corrections were made in the text.

- L132,133 Please consider changing to “contingency table” and “chi-squared test for independence”

Answer: : We would like to thank you for the comment. Corrections were made in the text.

- Figure 5 repeats the results from Table 2. One should be removed.

Answer: We would like to thank you for the comment. The manuscript has been corrected.

- Throughout the whole discussion, please give the exact drill names and analysed rpm, not drill D2 and speed 3.

Answer: We would like to thank you for the comment. The manuscript has been corrected.

- L164 Please rephase - Results are analyzed and presented in vc, not in terms of rpm.

Answer: We would like to thank you for the comment. The manuscript has been corrected. The results are presented in cutting speed.

- L172 adjoining? Did you mean adjacent?

Answer: We would like to thank you for the comment. The manuscript has been corrected.

- L175 vital bone? Did you mean intact?

Answer: We would like to thank you for the comment. The manuscript has been corrected.

- L179-180 not clear. You mean geometric growth (exponential growth) or the effect of drill geometry (effect of drill radius on tangential speed)? Please rephase.

Answer: We would like to thank you for the comment. The manuscript has been corrected with the “effect of drill geometry (effect of drill radius on tangential speed)”.

- L189-190 Neither hole diameter nor insertion torque of sample implant was measured. Please rephase or remove. Moreover, as you stated, undersized drilling is an intentional drilling for improving implant stability.

Answer: We would like to thank you for the comment. As rightly noted, we state that we did not evaluate the diameter of the holes obtained and we did not insert implants into them, so we also did not measure the torque value during implantation.

- L191 strong?

Answer: We would like to thank you for the comment. The manuscript has been corrected.

- L191-192 Reference for this review needed.

Answer: We would like to thank you for the comment. The manuscript has been corrected.

- L200 underway for many years? Did you mean “will last for many years”?

Answer: We would like to thank you for the comment. The manuscript has been corrected

- L204 Which process? Compression? Please rephase.

Answer: We would like to thank you for the comment. The manuscript has been corrected.

- L213 Please change “melted zones” to “carbonated areas”.

Answer: We would like to thank you for the comment. The manuscript has been corrected.

- L213 compact? Did you mean dense?

Answer: We would like to thank you for the comment. The manuscript has been corrected.

- L212-214 Presence of osteocytes was not measured, please rephase.

Answer: In this method, the osteocytes number calculation is not possible. We used the auto fluorescent properties to observe the bone surface not the bone cells. The magnification used and area study is making that calculation of osteocytes presence senseless.

Reviewer 4 Report

The submitted paper deals with the impact of different implant site preparation system on bone integrity. Though the paper is generally well written and readable, it reports only a qualitative evaluation made by the authors on the base of fluorescence microscopic analysis of the bone specimens. Although the authors adopted a scoring system, this seems to be arbitrary and does not have a term of comparison in the literature.  Moreover, it is not explained whether the bone alterations observed can clinically affect osseointegration. In other words, it is not clear whether the impact of different implant site preparation systems have a clinical relevance after all.

The authors are encouraged to enrich their research with further and more accurate analysis, e.g. histological analysis on fresh bone blocks to validate and supplement these first observations.

Author Response

Dear Reviewer,

We would like to express our sincerest gratitude to the Reviewers for their enormous efforts in criticizing the manuscript. We have taken into account all raised question here follows the detailed answers to the Reviewers.

REVIEWER#4

The submitted paper deals with the impact of different implant site preparation system on bone integrity. Though the paper is generally well written and readable, it reports only a qualitative evaluation made by the authors on the base of fluorescence microscopic analysis of the bone specimens. Although the authors adopted a scoring system, this seems to be arbitrary and does not have a term of comparison in the literature. Moreover, it is not explained whether the bone alterations observed can clinically affect osseointegration. In other words, it is not clear whether the impact of different implant site preparation systems have a clinical relevance after all.

The authors are encouraged to enrich their research with further and more accurate analysis, e.g. histological analysis on fresh bone blocks to validate and supplement these first observations.

Answer: We would like to thank you for the comment. The manuscript has been extended according to reviewer’s indications.

Round 2

Reviewer 1 Report

"Cortical thickness of the ribs was around 1mm and the periosteum was entirely removed."

"The drills had a diameter of 2-3mm. The ribs used for this scientific work had 1mm cortical bone thickness." "For the preparation procedure, drill bits were used in the sequence to prepare the bone bed for the smallest diameter of implant for a given system ( Osstem - 3,0 mm, Straumann - 3,3 mm, S-wide - 5,5 mm)."

  • rib cortical data is repetitive
  • drills had 2-3 mm but implant in case of S-Wide was 5.5mm. This should be clariefied, which drills prepared actually the holes for the Figures?

"The axial pressure values during drillings ranged from 0 to 720 g (750 ± 12 g)."

- I think mean value should be within minimal and maximal values.

In table 2. some of the cells contain value of zero, but Chi-squared test was conducted. Is it true? Chi-squared test is not applicable (maybe only with some correction e.g. Altman) if cells contain values less than 5. Or Fisher's exact test is more appropriate. I think some concerns should be raised even (and discussed), if the upper value of 95%CI is around 634!

Author Response

Dear Reviewer,

We would like to express our sincerest gratitude to the Reviewers for their enormous efforts in criticizing the manuscript. Their comments concerning statistical analysis were justified and helped to enhance this scientific work. We have taken into account all raised question here follows the detailed answers to the Reviewers.

  1. Question:

"Cortical thickness of the ribs was around 1mm and the periosteum was entirely removed."

"The drills had a diameter of 2-3mm. The ribs used for this scientific work had 1mm cortical bone thickness." "For the preparation procedure, drill bits were used in the sequence to prepare the bone bed for the smallest diameter of implant for a given system ( Osstem - 3,0 mm, Straumann - 3,3 mm, S-wide - 5,5 mm)."

rib cortical data is repetitive

Answer: We would like to thank you for the comment. The manuscript has been corrected.

     2. Question:

drills had 2-3 mm but implant in case of S-Wide was 5.5mm. This should be clariefied, which drills prepared actually the holes for the Figures?

Answer: We would like to thank you for the comment. S-Wide implants are short and wide. In the case of these implants, dedicated drills were used to achieve a diameter up to a 5.5 mm. We wanted to compare these implants with classic implants.

      3. Question:

"The axial pressure values during drillings ranged from 0 to 720 g (750 ± 12 g)."

- I think mean value should be within minimal and maximal values.

Answer: We would like to thank you for the comment. The aim of the experiment was to reproduce the clinical conditions as closely as possible, therefore the mean pressure force cannot be precisely determined, and the preparation is carried out in vitro always within the range of variable forces.

    4. Question:

In table 2. some of the cells contain value of zero, but Chi-squared test was conducted. Is it true? Chi-squared test is not applicable (maybe only with some correction e.g. Altman) if cells contain values less than 5. Or Fisher's exact test is more appropriate. I think some concerns should be raised even (and discussed), if the upper value of 95%CI is around 634!

Answer: We would like to thank you for the comment. Answer regarding Question 4 from the #1 Reviewer, 2nd and 3rd question from #3 Reviewer:

Because all of the Reviewer comments concerning statistical analysis were justified, we performed statistical analysis of obtained data once again, by another statistician and we corrected description of results:

- We changed data in Table 3. Really, data concerning variable “cutting speed” should not be analyzed for OR calculation, because there were 100% of true positive results for “<22 m/s” in group “OK”.

- We changed description of results in Results section.

- We changed title of Table 2 on more adequate.

- We added paragraph “Statistical analysis” to Materials and Methods section, in which we described statistical methods used for data analysis.

Reviewer 2 Report

Authors made excellent job addressing the reviewers notes 

Author Response

Dear Reviewer,

We would like to express our sincerest gratitude to the Reviewers for their enormous efforts in criticizing the manuscript. Their comments concerning statistical analysis were justified and helped to enhance this scientific work. We have taken into account all raised question here follows the detailed answers to the Reviewers.

Authors made excellent job addressing the reviewers notes.

Reviewer 3 Report

Authors have made some corrections, but most of my main concerns remained unanswered.

  1. If you use tangential speed in your analyses and this value is a critical threshold value in statistical analysis, please give tangential speed for each drill/rpm in table.
  2. The information about all necessary operation to raw data required for performing proper statistical analysis (like multiplication by 10 values in chi-square test) should be given in tables footnotes. Also the information that both Pearson’s chi-square test and Likehood Ratio chi-squared test were used in necessary (which test for which dataset).
  3. You can’t just “remove” 0 value from OR calculation !! Value 0 mean zero probability and OR is defined as result of dividing one probability by another, and you just can’t divide by zero ! calculation procedure of OR for cutting speed must be explained
  4. L94 Please correct physiological salt to physiological saline
  5. As osteocyte number was not measured (I do agree it was “senseless”, that’s why I made this comment, asking for rephrasing) therefore this fragment should be removed form the conclusion section.

Author Response

Dear Reviewer,

We would like to express our sincerest gratitude to the Reviewers for their enormous efforts in criticizing the manuscript. Their comments concerning statistical analysis were justified and helped to enhance this scientific work. We have taken into account all raised question here follows the detailed answers to the Reviewers.

  1. Question: If you use tangential speed in your analyses and this value is a critical threshold value in statistical analysis, please give tangential speed for each drill/rpm in table.

Answer: We would like to thank you for the comment. Specifying the speed ranges in units of revolutions / minute is connected with the necessity to standardize the revolutions for all drills. For statistical purposes, tangential speed in meters / second is used due to the difference in drill diameter, especially in the S-wide system. Therefore, in Table 1, the speeds are given in meters / second.

      2. Question: The information about all necessary operation to raw data required for performing proper statistical analysis (like multiplication by 10 values in chi-square test) should be given in tables footnotes. Also the information that both Pearson’s chi-square test and Likehood Ratio chi-squared test were used in necessary (which test for which dataset).

Answer: We would like to thank you for the comment. Answer regarding Question 4 from the #1 Reviewer, 2nd and 3rd Question from #3 Reviewer:

Because all of the Reviewer comments concerning statistical analysis were justified, we performed statistical analysis of obtained data once again, by another statistician and we corrected description of results:

- We changed data in Table 3. According to Authors, data concerning variable “cutting speed” should not be analyzed for OR calculation, because there were 100% of true positive results for “<22 m/s” in group “OK”.

- We changed description of results in Results section.

- We changed title of Table 2 on more adequate.

- We added paragraph “Statistical analysis” to Materials and Methods section, in which we described statistical methods used for data analysis.

     3. Question: You can’t just “remove” 0 value from OR calculation !! Value 0 mean zero probability and OR is defined as result of dividing one probability by another, and you just can’t divide by zero ! calculation procedure of OR for cutting speed must be explained.

Answer: We would like to thank you for the comment. Answer regarding Question 4 from the #1 Reviewer, 2nd and 3rd Question from #3 Reviewer:

Because all of the Reviewer comments concerning statistical analysis were justified, we performed statistical analysis of obtained data once again, by another statistician and we corrected description of results:

- We changed data in Table 3. Really, data concerning variable “cutting speed” should not be analyzed for OR calculation, because there were 100% of true positive results for “<22 m/s” in group “OK”.

- We changed description of results in Results section.

- We changed title of Table 2 on more adequate.

- We added paragraph “Statistical analysis” to Materials and Methods section, in which we described statistical methods used for data analysis.

   4. Question: L94 Please correct physiological salt to physiological saline.

Answer: We would like to thank you for the comment. The manuscript has been corrected.

   5. As osteocyte number was not measured (I do agree it was “senseless”, that’s why I made this comment, asking for rephrasing) therefore this fragment should be removed form the conclusion section.

Answer: We would like to thank you for the comment. The manuscript has been corrected.

Reviewer 4 Report

The authors have addressed all the points raised by reviewers. However, since I suggested to reject the manuscript in the first instance, I leave the final choice to the other reviewers, who indicated a major revision, and to the Editor.

Author Response

Dear Reviewer,

We would like to express our sincerest gratitude to the Reviewers for their enormous efforts in criticizing the manuscript. Their comments concerning statistical analysis were justified and helped to enhance this scientific work. We have taken into account all raised question here follows the detailed answers to the Reviewers.

The authors have addressed all the points raised by reviewers. However, since I suggested to reject the manuscript in the first instance, I leave the final choice to the other reviewers, who indicated a major revision, and to the Editor.

Round 3

Reviewer 1 Report

-

Author Response

Dear Reviewer,

We would like to express our sincerest gratitude to the Reviewers for their enormous efforts in criticizing the manuscript. The comments concerning especially statistical analysis were justified and helped to enhance this scientific work. All remarks have been included in the revised version of the manuscript. The last statistical remark has been also applied to the article.

Reviewer 3 Report

I have no more comments.

Author Response

(The authors gave the same response as above.)
